# ATOM3D:
# Tasks On Molecules in Three Dimensions

## Abstract

While a variety of methods have been developed for predicting molecular properties, deep learning networks that operate directly on three-dimensional molecular structure have recently demonstrated particular promise. In this work we present ATOM3D, a collection of both novel and existing datasets spanning several key classes of biomolecules, to systematically assess such learning methods. We develop three-dimensional molecular learning networks for each of these tasks, finding that they consistently improve performance relative to one- and two-dimensional methods. The specific choice of architecture proves to be critical for performance, with three-dimensional convolutional networks excelling at tasks involving complex geometries, while graph networks perform well on systems requiring detailed positional information. Furthermore, equivariant networks show significant promise but are currently unable to scale. Our results indicate many molecular problems stand to gain from three-dimensional molecular learning. All code and datasets are available at `github.com/xxxxxxx/xxxxxx`.

## 1 Introduction

A molecule's three-dimensional (3D) shape is critical to understanding its physical mechanisms of action, and can be used to answer a number of questions relating to drug discovery, molecular design, and fundamental biology. A molecule's atoms often adopt specific 3D configurations that minimize its free energy, and by representing these 3D positions—the atomistic geometry—we can model this 3D shape in ways that would not be possible with 1D or 2D representations such as linear sequences or chemical bond graphs (Table 1). However, existing works that examine diverse molecular tasks, such as MoleculeNet (Wu et al., 2018) or TAPE (Rao et al., 2019), focus on these lower dimensional representations. In this work, we demonstrate the benefit yielded by learning on 3D atomistic geometry and promote the development of 3D molecular learning by providing a collection of datasets leveraging this representation.

Furthermore, we argue that the atom should be considered a "machine learning datatype" in its own right, deserving focused study much like images in computer vision or text in natural language processing. All molecules, including proteins, small molecule compounds, and nucleic acids, can be homogeneously represented as atoms in 3D space. These atoms can only belong to a fixed class of element types (e.g. carbon, nitrogen, oxygen), and are all governed by the same underlying laws of physics, leading to important rotational, translational, and permutational symmetries. These systems also contain higher-level patterns that are poorly characterized, creating a ripe opportunity for learning them from data: though certain basic components are well understood (e.g. amino acids, nucleic acids, functional groups), many others can not easily be defined. These patterns are in turn composed in a hierarchy that itself is only partially elucidated.

While deep learning methods such as graph neural networks (GNNs) and convolutional neural networks (CNNs) seem especially well suited to atomistic geometry, to date there has been no systematic evaluation of such methods on molecular tasks. Additionally, despite the growing number of 3D structures available in databases such as the Protein Data Bank (PDB) (Berman et al., 2000), they require significant processing before they are useful for machine learning tasks. Inspired by the success of accessible databases such as ImageNet (Jia Deng et al., 2009) and SQuAD (Rajpurkar et al., 2016) in sparking progress in their respective fields, we create and curate benchmark datasets for atomistic tasks, process them into a simple and standardized format, systematically benchmark

Table 1: Representation choice for molecules. Adding in 3D information consistently improves performance. The depicted 1D representations are the amino acid sequence and SMILES (Weininger, 1988) for proteins and small molecules, respectively.

| Structure Level | Dimension | Representation | Examples | |
|---|---|---|---|---|
| | | | Proteins | Small Molecules |
| primary | 1D | linear sequence | KVKALPDA | CC(C)CC(C)NO |
| secondary | 2D | chemical bond graph | | |
| tertiary | 3D | atomistic geometry | | |

3D molecular learning methods, and present a set of best practices for other machine learning researchers interested in entering the field of 3D molecular learning. We develop new methods for several datasets and reveal a number of insights related to 3D molecular learning, including the consistent improvements yielded by using atomistic geometry, the lack of a single dominant method, and the presence of several tasks that can be improved through 3D molecular learning.

## 2 RELATED WORK

Three dimensional molecular data have long been pursued as an attractive source of information in molecular learning and chemoinformatics, but until recently have achieved underwhelming results relative to 1D and 2D representations (Swamidass et al., 2005; Azencott et al., 2007). However, due to increases in data availability and methodological advances, machine learning methods based on 3D molecular structure have begun to demonstrate significant impact in the last couple of years on specific tasks such as protein structure prediction (Senior et al., 2020), equilibrium state sampling (Noé et al., 2019), and drug design (Zhavoronkov et al., 2019). While there have been some broader assessments of groups of related biological tasks, these have focused on on either 1D (Rao et al., 2019) or 2D (Wu et al., 2018) representations. By focusing instead on atomistic geometry, we can consistently improve performance and address disparate problems involving any combination of small molecules, proteins, and nucleic acids through a unified lens.

Graph neural networks (GNNs) have grown to be a major area of study, providing a natural way of learning from data with complex spatial structure. Many GNN implementations have been motivated by applications to atomic systems, including molecular fingerprinting (Duvenaud et al., 2015), property prediction (Schütt et al., 2017; Gilmer et al., 2017; Liu et al., 2019), protein interface prediction (Fout et al., 2017), and protein design (Ingraham et al., 2019). Instead of encoding points in Euclidean space, GNNs encode their pairwise connectivity, capturing a structured representation of atomistic data.

Three-dimensional CNNs (3DCNNs) have also become popular as a way to capture these complex 3D geometries. They have been applied to a number of biomolecular applications such as protein interface prediction (Townshend et al., 2019), protein model quality assessment (Pagès et al., 2019; Derevyanko et al., 2018), protein sequence design (Anand et al., 2020), and structure-based drug discovery (Wallach et al., 2015; Torng & Altman, 2017; Ragoza et al., 2017; Jiménez et al., 2018). These 3DCNNs can encode translational and permutational symmetries, but incur significant computational expense and cannot capture rotational symmetries without data augmentation.

In an attempt to address many of the problems of representing atomistic geometries, equivariant neural networks (ENNs) have emerged as a new class of methods for learning from molecular systems. These networks are built such that geometric transformations of their inputs lead to well-defined transformations of their outputs. This setup leads to the neurons of the network learning rules that resemble physical interactions. Tensor field networks (Thomas et al., 2018) and Cormorant (Kondor,

2018; Anderson et al., 2019) have applied these principles to atomic systems and begun to demonstrate promise on extended systems (Eismann et al., 2020; Weiler et al., 2018). However, in general, these methods have not been applied to larger-scale molecular tasks.

## 3   3D MOLECULAR LEARNING

We define 3D molecular learning as the set of tasks where the input space is atoms in three dimensions. We write this space as $\mathbb{A}^N$ where $\mathbb{A} = \mathbb{P} \times \mathbb{E}$. $\mathbb{P} = \mathbb{R}^3$ is the position space and $\mathbb{E} = \{C, H, O, N, P, S, ...\}$ is the element space.

We select 3D molecular learning tasks from structural biophysics and medicinal chemistry that span a variety of molecule types and address a range of important problems. Multiple of these datasets are novel, while others are extracted from existing sources (Table 2). We provide all datasets in a standardized format that requires no specialized libraries. Alongside these datasets, we present corresponding best practices, including splitting and filtering criteria, to minimize data leakage concerns and ensure generalizability and reproducibility. Taken together, we hope these efforts will lower the barrier to entry for machine learning researchers interested in developing methods for 3D molecular learning and encourage rapid progress in the field. Detailed descriptions of the preparation of each dataset can be found in Appendix C.1.

Table 2: Tasks included in ATOM3D dataset, along with schematic representation of their inputs. P indicates protein, SM indicates small molecule, R indicates RNA. Lines indicate interaction and the smaller square within proteins indicates an individual amino acid. New datasets are in bold.

| Name (Task Code) | Schematic | Objective | Source |
|---|---|---|---|
| Small Molecule Properties (SMP) | SM | Properties | QM9 (Ruddigkeit et al., 2012) |
| Protein Interface Prediction (PIP) | P1 — P2 | Amino Acid Interaction | DIPS (Townshend et al., 2019) DB5 (Vreven et al., 2015) |
| **Residue Identity (RES)** | P | **Amino Acid Identity** | **New, created from PDB (Berman et al., 2000)** |
| **Mutation Stability Prediction (MSP)** | P1 — P2 vs. P1 — P2 | **Effect of Mutation** | **New, created from SKEMPI (Jankauskaitė et al., 2019)** |
| Ligand Binding Affinity (LBA) | P — SM | Binding Strength | PDBBind (Wang et al., 2004) |
| **Ligand Efficacy Prediction (LEP)** | P — SM vs. P — SM | **Drug Efficacy** | **New, created from PDB (Berman et al., 2000)** |
| Protein Structure Ranking (PSR) | P | Ranking | CASP-QA (Kryshtafovych et al., 2019) |
| RNA Structure Ranking (RSR) | R | Ranking | FARFAR2-Puzzles (Watkins & Das, 2019) |

### 3.1 SMALL MOLECULE PROPERTIES (SMP)

**Impact** – Predicting physico-chemical properties of small molecules is a common task in medicinal chemistry and materials design. Quantum chemical calculations can save expensive experiments but are themselves costly and cannot cover the huge chemical space spanned by candidate molecules.
**Dataset** – The QM9 dataset (Ruddigkeit et al., 2012; Ramakrishnan et al., 2014) contains structures and energetic, electronic, and thermodynamic properties for 134,000 stable small organic molecules, obtained from quantum-chemical calculations.
**Metrics** – We predict the molecular properties from the ground-state structure.
**Split** – We split molecules randomly.

### 3.2 PROTEIN INTERFACE PREDICTION (PIP)

**Impact** – Proteins interact with each other in many scenarios—for example, antibody proteins recognize diseases by binding to antigens. A critical problem in understanding these interactions is to identify which amino acids of two given proteins will interact upon binding.
**Dataset** – For training, we use the Database of Interacting Protein Structures (DIPS), a comprehensive dataset of protein complexes mined from the PDB (Townshend et al., 2019). We predict on the Docking Benchmark 5 (Vreven et al., 2015), a smaller gold standard dataset.
**Metrics** – We predict if two amino acids will come into contact when their respective proteins bind.
**Split** – We split protein complexes by sequence identity at 30%.

### 3.3 RESIDUE IDENTITY (RES)

**Impact** – Understanding the structural role of individual amino acids is important for engineering new proteins. We can understand this role by predicting the propensity for different amino acids at a given protein site based on the surrounding structural environment (Torng & Altman, 2017).
**Dataset** – We generate a novel dataset consisting of atomic environments extracted from non-redundant structures in the PDB.
**Metrics** – We formulate this as a classification task where we predict the identity of the amino acid in the center of the environment based on all other atoms.
**Split** – We split residue environments by protein topology class.

### 3.4 MUTATION STABILITY PREDICTION (MSP)

**Impact** – Identifying mutations that stabilize a protein's interactions is a key task in designing new proteins. Experimental techniques for probing these are labor-intensive (Antikainen & Martin, 2005; Lefèvre et al., 1997), motivating the development of efficient computational methods.
**Dataset** – We derive a novel dataset by collecting single-point mutations from the SKEMPI database (Jankauskaitė et al., 2019) and model each mutation into the structure to produce mutated structures.
**Metrics** – We formulate this as a binary classification task where we predict whether the stability of the complex increases as a result of the mutation.
**Split** – We split protein complexes by sequence identity at 30%.

### 3.5 LIGAND BINDING AFFINITY (LBA)

**Impact** – Most therapeutic drugs and many molecules critical for biological signaling take the form of small molecules. Predicting the strength of the protein-small molecule interaction is a challenging but crucial task for drug discovery applications.
**Dataset** – We use the PDBBind database (Wang et al., 2004; Liu et al., 2015), a curated database containing protein-ligand complexes from the PDB and their corresponding binding strengths.
**Metrics** – We predict $pK = -\log(K)$, where $K$ is the binding affinity in Molar units.
**Split** – We split protein-ligand complexes by protein sequence identity at 30%.

### 3.6 LIGAND EFFICACY PREDICTION (LEP)

**Impact** – Many proteins switch on or off their function by changing shape. Predicting which shape a drug will favor is thus an important task in drug design.

**Dataset** – We develop a novel dataset by curating proteins from several families with both "active" and "inactive" state structures, and model in 527 small molecules with known activating or inactivating function using the program Glide (Friesner et al., 2004).

**Metrics** – We formulate this as a binary classification task where we predict whether or not a molecule bound to the structures will be an activator of the protein's function or not.

**Split** – We split complex pairs by protein.

### 3.7 PROTEIN STRUCTURE RANKING (PSR)

**Impact** – Proteins are one of the primary workhorses of the cell, and knowing their structure is often critical to understanding (and engineering) their function.

**Dataset** – The Critical Assessment of Structure Prediction (CASP) (Kryshtafovych et al., 2019) is a blind international competition for predicting protein structure.

**Metrics** – We formulate this as a regression task, where we predict the global distance test (GDT_TS) from the true structure for each of the predicted structures submitted in the last 18 years of CASP.

**Split** – We split structures temporally by competition year.

### 3.8 RNA STRUCTURE RANKING (RSR)

**Impact** – Similar to proteins, RNA plays major functional roles (e.g., gene regulation) and can adopt well-defined 3D shapes. Yet the problem is data-poor, with only a few hundred known structures.

**Dataset** – Candidate models generated by FARFAR2 (Watkins & Das, 2019) for the first 21 released RNA Puzzle challenges (Cruz et al., 2012), a blind structure prediction competition for RNA.

**Metrics** – We predict the root-mean-squared deviation (RMSD) from the ground truth structure.

**Split** – We split structures temporally by competition year.

## 4 EXPERIMENTAL SETUP

To assess the benefits of 3D molecular learning, we use a combination of existing and novel 3D molecular learning methods, and implement a number of robust baselines. Our 3D molecular learning methods belong to one of each of the major classes of deep learning algorithms that have been applied to atomistic systems: graph networks, three-dimensional convolutional networks, and equivariant networks. Here we describe the core networks and the novel extensions needed to adapt them to certain datasets. See Appendix C.2 for task-specific details and hyperparameters.

### 4.1 CORE NETWORKS

For GNNs, we represent molecular systems as graphs in which each node is an atom. Edges are defined between all atoms separated by less than 4.5 Å, and weighted by the distance between the atoms. Node features are one-hot-encoded by atom type. Our core model uses five layers of graph convolutions, each followed by batch normalization and ReLU activation, a sum pooling layer, and two fully-connected layers with dropout.

For 3DCNNs, we represent our data as a cube of fixed size (different per task due to the different molecular sizes) in 3D space that is discretized into voxels with resolution of 1 Å to form a grid. Each voxel is associated with a one-hot-encoded vector which denotes the presence or absence of each atom type. Our core model consists of four 3D-convolutional layers, each followed by maxpooling, dropout, and ReLU activation, and two fully-connected layers.

For ENNs, we use SE(3)-equivariant networks that represent each atom of a structure by its position as absolute coordinates in 3D space with one-hot-encoded atom type as features. No rotational augmentation is needed due to the rotational symmetry of the network. The core of all architectures in this work is a network of four layers of covariant neurons that use the Clebsch–Gordan transform as nonlinearity, as described and implemented in Anderson et al. (2019).

### 4.2 SIAMESE ARCHITECTURES

For tasks involving comparing two sets of atoms sampled from the same distributions, we also develop new architectures that are Siamese in nature. Specifically, the PIP dataset involves predicting

a symmetric interaction between two proteins, while the MSP and LEP datasets involve a symmetric comparison between two interactions. Taking inspiration from Townshend et al. (2019)'s use of a Siamese 3DCNN network for the PIP dataset, we replicate that architecture for our PIP, MSP, and LEP datasets, and develop new Siamese GNN and Siamese ENN networks. Specifically, we train a pair of core networks with tied weights, ensuring symmetric treatment of both items of the pair. We then combine the final learned embeddings from both core networks to output a final prediction. Beyond the novelty of weight-tying atom-level GNNs and ENNs, to our knowledge this is also the first use of weight-tying across SE(3)-equivariant networks.

### 4.3 AMINO ACID OUTPUTS

Certain tasks involve making a prediction on a specific amino acid (PIP, RES, and MSP; see Table 2), yet GNNs and ENNs typically rely on summing over all node embeddings to compute a final graph embedding, making it difficult to isolate this amino acid. To remedy this, after our convolutional layers we implement the novel procedure of extracting the embedding of only the C$\alpha$ atom of the amino acid in question, thereby allowing our GNNs and ENNs to isolate it.

## 5 RESULTS

To assess the utility of 3D molecular learning, we evaluate our methods on the ATOM3D datasets and compare performance to state-of-the-art methods using 1D or 2D representations. We stress that in many cases, 3D molecular learning methods have never been applied to the proposed tasks, and that several of the tasks are novel. In the following sections, we describe the results of our benchmarking and some key insights that can be derived from them. We also aggregate these results along with additional metrics and standard deviations over three replicates in Appendix E.

### 5.1 3D REPRESENTATIONS CONSISTENTLY IMPROVE PERFORMANCE

Our evaluation of 3D methods on the tasks in ATOM3D reveals that incorporating atomistic geometry leads to consistently superior performance compared to 1D and 2D methods. For small molecules, state-of-the-art methods do not use 1D representations, so we focus instead on comparing to representations at the 2D level, i.e. the chemical bond graph. This is the approach taken by the 2D GNN introduced by Tsubaki et al. (2019) or the N-gram graph method by Liu et al. (2019), which both obtain similar results (Table 3) on the small-molecule-only dataset SMP. When we add 3D distance, as done for our GNN model, we improve performance across all targets in SMP (Table 3).

For tasks involving biopolymers (proteins and RNA), state-of-the-art methods do not use 2D representations, primarily because most of the chemical bond graph can be easily re-derived from the 1D representation, i.e. the linear sequence that makes up the biopolymer. We thus compare to representations at the 1D level (Table 7). For MSP and RES, both new datasets, we evaluate against Rao et al. (2019)'s TAPE model, a transformer architecture that operates on protein sequence and is state-of-the-art amongst 1D methods for many tasks. For PIP, we compare to the sequence-only version of BIPSPI (Sanchez-Garcia et al., 2018), a state-of-the-art boosted decision tree method for protein interaction prediction. We find that 3D methods outperform these 1D methods on all biopolymer-only datasets (PIP, RES, MSP).

For tasks involving both biopolymers and small molecules, we compare DeepDTA (Öztürk et al., 2018). This network uses a 1D representation via a 1DCNN for both the biopolymer and small molecules. For LBA, we additionally compare to DeepAffinity (Karimi et al., 2019) which uses pairs of a ligand SMILES string and a novel representation of structurally-annotated protein sequences. Using a 3D representation for both ligand and protein instead leads to improved performance for the joint protein-small molecule datasets (LBA and LEP, see Table 5).

The biopolymer structure ranking tasks (PSR and RSR) are inherently 3D in nature, as they involve evaluating the correctness of different 3D shapes taken on by the same biopolymer. Thus, critically, a 1D or 2D representation would not be able to differentiate between these different shapes since the linear sequence and chemical bond graph would remain the same. We therefore compare to state-of-the-art 3D methods as shown in Table 6.

Table 3: Small molecule results. Metric is mean absolute error (MAE).

| Task | Target | 3D | | | Non-3D | |
|------|--------|------|-----|-----|--------------------|-----------------|
| | | 3DCNN | GNN | ENN | Tsubaki et al. (2019) | Liu et al. (2019) |
| SMP | $\mu$ [D] | 0.572 | 0.068 | **0.046** | 0.496 | 0.520 |
| | $\varepsilon_{\text{gap}}$ [eV] | 0.589 | 0.091 | **0.065** | 0.154 | 0.184 |
| | $U_0^{\text{at}}$ [eV] | 1.615 | 0.070 | **0.023** | 0.182 | 0.218 |

Table 4: Biopolymer results. AUROC is the area under the receiver operating characteristic curve. Asterisks (*) indicate that the exact training data differed (though splitting criteria were the same).

| Task | Metric | 3D | | | Non-3D |
|------|--------|------|-----|-----|--------------------------|
| | | 3DCNN | GNN | ENN | Sanchez-Garcia et al. (2018) |
| PIP | AUROC | **0.844** | *0.669 | — | 0.841 |
| | | | | | Rao et al. (2019) |
| RES | accuracy | **0.451** | 0.082 | *0.072 | *0.30 |
| MSP | AUROC | 0.520 | 0.637 | **0.678** | 0.554 |

Table 5: Joint small molecule/biopolymer results. $R_S$ is Spearman correlation, $R_P$ is Pearson correlation, AUROC is area under the receiver operating characteristic curve, and RMSD is root-mean-squared deviation. Asterisks (*) indicate that the exact training data differed (though splitting criteria were the same).

| Task | Metric | 3D | | | Non-3D | |
|------|--------|------|-----|-----|------------------|------------------|
| | | 3DCNN | GNN | ENN | Öztürk et al. (2018) | Karimi et al. (2019) |
| LBA | RMSD | 1.520 | 1.936 | ***1.429** | 1.565 | 1.893 |
| | glob. $R_P$ | 0.558 | **0.581** | *0.541 | 0.573 | 0.415 |
| | glob. $R_S$ | 0.556 | **0.647** | *0.532 | 0.574 | 0.426 |
| LEP | AUROC | **0.824** | 0.678 | 0.569 | 0.696 | — |

Table 6: Structure ranking results. $R_S$ is Spearman correlation, $R_P$ is Pearson correlation. Mean measures the correlation for structures corresponding to the same biopolymer, whereas global measures the correlation across all biopolymers.

| Task | Metric | 3D | | |
|------|--------|------|-----|------------------------|
| | | 3DCNN | GNN | SotA |
| PSR | mean $R_S$ | 0.177 | 0.327 | **0.432** (Pagès et al., 2019) |
| | glob. $R_S$ | **0.837** | 0.716 | 0.796 (Pagès et al., 2019) |
| RSR | mean $R_S$ | **0.414** | 0.195 | 0.173 (Alford et al., 2017) |
| | glob. $R_S$ | **0.656** | 0.309 | 0.304 (Alford et al., 2017) |

More generally, we find that learning methods that leverage the 3D geometry of molecules hold state-of-the-art on all tasks on our benchmark (Appendix D).

## 5.2 MANY 3D MOLECULAR LEARNING PROBLEMS REMAIN UNDEREXPLORED

As demonstrated in the previous section, formulating a molecular problem through the lens of 3D molecular learning can lead to significantly improved performance. However, many important problems have not been studied within this framework, leaving significant room for further improvement. This opens up a ripe field of research with much low-hanging fruit. One prominent example we explore here is RNA structure ranking, where the state-of-the-art method uses Rosetta (Alford et al., 2017), a hand-designed potential energy function. When we instead apply our 3DCNN method that learns directly from the 3D atomistic geometry, we see dramatic increases in performance (Table 6).

In a similar vein, on the ligand efficacy prediction task we find that the 3DCNN method outperforms Glide (Friesner et al., 2004), a state-of-the-art scoring function for protein-small molecule docking. The 3DCNN achieves an AUROC of 0.824, compared to Glide's 0.770.

We also find room for improvement in domains where 3D molecular learning is already being employed. Protein structure ranking is one such area, and we see that the 3DCNN model is competitive with the state-of-the-art deep learning method by Pagès et al. (2019) (Table 6), surpassing it in terms of absolute assessment of correctness (i.e., comparing 3D candidates from different biopolymers) though not in terms of relative assessment (i.e., comparing 3D candidates from the same biopolymer).

Overall, these results demonstrate the potential of 3D molecular learning to address a wide range of problems involving molecular structure, and we anticipate that continued development of such models on less well-studied tasks will aid progress in biomedical research.

## 5.3 DIFFERENT TASKS REQUIRE DIFFERENT ARCHITECTURES

While atomistic methods consistently outperform their non-3D counterparts and provide a systematic way of representing molecular data, our results also provide evidence that architecture selection plays a critical role in performance. For tasks primarily focused on small molecules (SMP and LBA), we see superior performance from the particle-based methods (GNN and ENN) than from the volumetric 3DCNN. Small molecules are already intuitively represented as graph-like structures, with nodes (atoms) and edges (bonds), and the quantities computed in these small molecule tasks depend less on complex 3D geometry and more on the exact position of each atom relative to its neighbors. Unlike particle-based methods, 3DCNNs must approximate these positions, and while increasing spatial resolution increases precision, it also leads to cubic scaling of complexity.

On the other hand, for larger molecules and complex 3D geometries that are critical to tasks like PSR, RSR, LEP, and RES, we see that the 3DCNN method outperforms GNNs. Here, the 3DCNN's ability to directly represent differences between patterns in 3D space, as opposed to trying to reconstruct them through pairwise distances, is likely what allows it to perform well. In these tasks, the exact position and relational information between atoms is less important than their overall conformation.

Finally, equivariant networks show promise but suffer from scalability issues. One motivation for the development of equivariant networks is that they fill a "happy medium" where they can both represent atom positions precisely and capture complex geometries. On some of the tasks where we could test the ENN (LBA, SMP), we often observed close to state-of-the-art performance, even on a reduced training set (LBA). Unfortunately, current implementations do not yet scale to most of our tasks due to the compute- and memory-intensive nature of the Clebsch-Gordan products used to maintain rotational equivariance. For some tasks, the performance was severely limited by only training on a fraction of the data ($< 1\%$ for RES) or a portion of the entire atomic structure (LEP), and for others we could not apply the ENN at all. These limitations point to the need for further architectural innovations before their performance can be demonstrated on extended systems.

## 6 Conclusion

In this work we present a vision of 3D atom-level data as a new "machine learning datatype" deserving focused study. Atomistic data shares several underlying symmetries, contains poorly understood higher-level patterns, and can be used to address many high-impact but unsolved problems.

We create several benchmark datasets and compare the performance of different types of 3D molecular learning models across these tasks. Many of these architectures were developed specifically for the tasks in question, such as the Siamese ENN and GNN models used for paired tasks (PIP, MSP, and LEP). For tasks that can be formulated in lower dimensions, we demonstrate that 3D molecular learning yields consistent gains in performance over 1D and 2D methods. We also show that selection of an appropriate architecture is critical for optimal performance on a given task; depending on the structure of the underlying data, a 3DCNN, GNN, or ENN may be most appropriate. As equivariant networks continue to improve in efficiency and stability, we expect these to become more and more viable due to their close modeling of physical laws.

While ATOM3D establishes a first set of benchmark datasets, there are many other open areas in biomedical research and molecular science that are ripe for 3D molecular learning, especially as structural data becomes readily available. Such tasks include virtual screening and pose prediction of small molecule drugs, or the prediction of conformational ensembles instead of static structures. The use of multiple 3D conformations per molecule represents an especially promising direction, as they would more faithfully reproduce the entire set of states a given molecule could adopt. As such, we envision expanding the ATOM3D framework beyond the tasks described here.

Through this work, we hope to lower the entry barrier for machine learning practitioners, encourage the development of machine learning algorithms focused on 3D atomistic data, and promote a novel paradigm within the fields of structural biology and medicinal chemistry.

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
