# OpenReview forum: "ATOM3D: Tasks On Molecules in Three Dimensions"
_ICLR.cc/2021/Conference — Reject_

### Official Review · AnonReviewer4 · 2020-10-28
**An interesting resource but needs a bit more context**

**Rating:** 4
**Confidence:** 4

**Review:**

This paper presents a large benchmark of machine learning tasks for molecules represented by the 3D coordinates of their atoms. The benchmark is a combination of existing data sets and newly created ones, and covers a variety of applications and tasks, from small molecules to RNA or protein structures, and including classification, regression and ranking tasks. In addition, three deep-learning algorithms are implemented and evaluated on these benchmarks, and compared to state-of-the-art methods that do not use 3D information, and empirically demonstrate the benefit of incorporating 3D information in the networks.

The vast majority of machine learning methods that have been developed for molecules use either 1D or 2D information. Therefore, this resource and the empirical demonstration that using 3D information can improve performance can be very valuable to the community. However, I have a number of concerns about this paper:

1. I am assuming this has not been incorporated to the paper for anonymization reasons, but could the authors confirm that they are indeed planning to make both the data sets and the code used to produce the results (in particular, the three proposed neural networks architecture) available? This is obviously essential to the paper and I would feel more comfortable accepting a version of the paper that includes this information (possibly with URLs withdrawn if there is a concern about maintaining the review process blind).

2. The paper does not discuss the nature of the 3D information further than "By representing a molecule's atoms and their 3D positions". However, molecules do not have a fixed 3D structure, but rather multiple conformations driven in particular by rotatable bonds. Determining the multiple possible conformations of drug-like molecules is still an ongoing research topic (see for example the review of Hawkins (2017)), not to mention the determination of the 3D structure of proteins, which is indeed the topic of one of the data sets provided. What information is provided in the different data sets (a single conformation? multiple conformations?) and how does this affect both algorithms (if several conformations are used) and prediction performance?

3. I would really refrain from using "atomistic learning" to describe what the community has been referring to as "learning from 3D molecular representations" for decades.

Actually, the abstract (and, more generally, the paper) reads as if neural networks were the only kind of machine learning algorithms that could be applied to molecules and that very little work has been done in the past to incorporate 3D information in chemoinformatics. While it is true that most current techniques rely mostly on 2D (for small molecules) or 1D (for large molecules) representations, it is not for lack of trying to incorporate 3D information, but because 1) this information was either lacking or incomplete (in the sense that a single conformation gives somewhat limited information; for example, you may have the crystal structure of a protein, but that doesn't inform you directly on the pose of its pocket when binding a specific small molecule) and 2) earlier attempts at making use of 3D information have often found that it did not improve performance (see Swamidass et al. (2005) or Azencott et al. (2007)), either because of the aforementioned incompleteness or because the methods were not up to par. The framing of the paper ignores decades of work in chemoinformatics, in particular (but not limited to) around kernel methods. I am listing a few examples of such papers below, not because I think they should all be included in this paper, but because in my opinion the paper would benefit from considering this body of work.

In addition, although some authors have already used "atomistic machine learning" in the context of chemoinformatics (see Schütt et al. (2018)), the term "atomistic learning" is already often used in opposition to "holistic learning" in education.

4. Finally, in Section 4, the paper could benefit from stating very explicitly what is novel and what is not novel in the three proposed 3D architectures.

Axen, Seth D., et al. "A simple representation of three-dimensional molecular structure." Journal of medicinal chemistry 60.17 (2017): 7393-7409.

Azencott, C.-A., et al. "One-to four-dimensional kernels for virtual screening and the prediction of physical, chemical, and biological properties." Journal of chemical information and modeling 47.3 (2007): 965-974.

Gaüzere, B., Brun, L., and Villemin D,. "Two new graphs kernels in chemoinformatics." Pattern Recognition Letters 33.15 (2012): 2038-2047.

Hawkins, Paul C. D. "Conformation generation: the state of the art." Journal of Chemical Information and Modeling 57.8 (2017): 1747-1756.

Mahé, P., et al. "Graph kernels for molecular structure− activity relationship analysis with support vector machines." Journal of chemical information and modeling 45.4 (2005): 939-951.

Mohr, J. A., Jain, B. J., and Obermayer, K. "Molecule kernels: a descriptor-and alignment-free quantitative structure–activity relationship approach." Journal of chemical information and modeling 48.9 (2008): 1868-1881.

Nettles, J. H. et al. Bridging chemical and biological space: “target fishing” using 2D and 3D molecular descriptors. J. Medicinal Chem. 49, 6802–6810 (2006).

Rhodes, N., Clark, D. E. & Willett, P. Similarity searching in databases of flexible 3d structures using autocorrelation vectors derived from smoothed bounded distance matrices. J. Chem. Info. Mod. 46, 615–619 (2006).

Schütt, K. T. et al. SchNet - a deep learning architecture for molecules and materials. The Journal of Chemical Physics 148(24), 241722 (2018)

Swamidass, S. J., et al. "Kernels for small molecules and the prediction of mutagenicity, toxicity and anti-cancer activity." Bioinformatics 21.suppl_1 (2005): i359-i368.

---

> ### Author Response · Authors · 2020-11-24
> **Response to Reviewer 4**
>
> Thank you for your comments and especially for your detailed suggestions regarding contextualizing this work.  We respond to your individual concerns below.
>
> > I am assuming this has not been incorporated to the paper for anonymization reasons, but could the authors confirm that they are indeed planning to make both the data sets and the code used to produce the results (in particular, the three proposed neural networks architecture) available? This is obviously essential to the paper and I would feel more comfortable accepting a version of the paper that includes this information (possibly with URLs withdrawn if there is a concern about maintaining the review process blind).
>
> We agree that this is an essential part of this work, and we confirm we will be providing all code and datasets in a Github repository and on a dedicated website after the review process. In the meantime, please find anonymized versions of our code at this [link](https://drive.google.com/drive/folders/1UKqVvz9Ae35PThyqB7unlZNb2bT8d740?usp=sharing). We have also included a censored Github URL in the abstract, which will be replaced by the final URL in the final version of the manuscript.
>
> > The paper does not discuss the nature of the 3D information further than "By representing a molecule's atoms and their 3D positions". However, molecules do not have a fixed 3D structure, but rather multiple conformations driven in particular by rotatable bonds. Determining the multiple possible conformations of drug-like molecules is still an ongoing research topic (see for example the review of Hawkins (2017)), not to mention the determination of the 3D structure of proteins, which is indeed the topic of one of the data sets provided. What information is provided in the different data sets (a single conformation? multiple conformations?) and how does this affect both algorithms (if several conformations are used) and prediction performance?
>
> We currently focus on conformations where a molecule’s free energy is minimized. These conformations are typically approximated through wet-lab experiment, though they are occasionally computationally modeled instead as is the case for SMP, and parts of MSP and LEP. Molecules, and especially macromolecules, often have one or a few specific conformations that minimize their free energy, and they spend most of their time in those conformations, so we find we can get robust performance by focusing on those.  You are correct of course that considering a larger set of conformations could lead to improved performance—this represents an exciting avenue for future research!  We have added text discussing our current use of free energy minimum states in the introduction, and expand on this future avenue of research in the conclusion.
>
> In terms of our specific datasets, PSR and RSR each have many thousands of conformations per molecule, with the goal being to pick out the ones that most resemble the experimentally determined structure.  We tackle this through a pointwise approach, scoring one conformation at a time, but considering ensembles instead could likely further improve our performance (which is already quite strong on these tasks).  LEP and MSP involve comparing pairs of conformations, and in these cases we do input both conformations simultaneously into our predictors, as these tasks are specifically about comparing these conformations. PIP, LBA, RES, and SMP focus on a single conformation.

---

> > ### Author Response · Authors · 2020-11-24
> > **Response to Reviewer 4 - Continued**
> >
> > > I would really refrain from using "atomistic learning" to describe what the community has been referring to as "learning from 3D molecular representations" for decades. [...] In addition, although some authors have already used "atomistic machine learning" in the context of chemoinformatics (see Schütt et al. (2018)), the term "atomistic learning" is already often used in opposition to "holistic learning" in education.
> >
> > We were unaware of the use of “atomistic learning” in an education context; thank you for bringing this to our attention.  To reduce the ambiguity around the name, and to better place this work within the broader chemoinformatics context, we have changed all references to “atomistic learning” to “3D molecular learning”.
> >
> > > Finally, in Section 4, the paper could benefit from stating very explicitly what is novel and what is not novel in the three proposed 3D architectures.
> >
> > We have explicitly indicated the novel aspects for the deep learning methods presented in Section 4, and also replicate these aspects here.  The 3DCNN architectures are standard; the novelty lies in the GNN and ENN architectures.  First, the weight-tying across two parallel networks for our atom-level GNN and ENN architectures is new (in fact, we believe this to be the first implementation of weight-tying across SE(3)-equivariant networks).  Second, representing the GNN and ENN inputs at the atomic level and then extracting the embeddings at the alpha-carbon node to make amino-acid-specific predictions is novel.
> >
> > > Actually, the abstract (and, more generally, the paper) reads as if neural networks were the only kind of machine learning algorithms that could be applied to molecules and that very little work has been done in the past to incorporate 3D information in chemoinformatics. While it is true that most current techniques rely mostly on 2D (for small molecules) or 1D (for large molecules) representations, it is not for lack of trying to incorporate 3D information, but because 1) this information was either lacking or incomplete (in the sense that a single conformation gives somewhat limited information; for example, you may have the crystal structure of a protein, but that doesn't inform you directly on the pose of its pocket when binding a specific small molecule) and 2) earlier attempts at making use of 3D information have often found that it did not improve performance (see Swamidass et al. (2005) or Azencott et al. (2007)), either because of the aforementioned incompleteness or because the methods were not up to par. The framing of the paper ignores decades of work in chemoinformatics, in particular (but not limited to) around kernel methods. I am listing a few examples of such papers below, not because I think they should all be included in this paper, but because in my opinion the paper would benefit from considering this body of work.
> >
> > Thank you for pointing this out; this was certainly not our intention, and we have worked to clarify that deep learning methods are not the only way to incorporate 3D molecular information.  Specifically, we have re-worked the abstract and updated the related work section to include references to previous work around 3D and kernel methods in chemoinformatics.

---

### Official Review · AnonReviewer3 · 2020-10-28
**Review for: ATOM3D: Tasks On Molecules in Three Dimensions**

**Rating:** 6
**Confidence:** 3

**Review:**

In this paper, the authors introduce a repository of datasets for several atomistic learning tasks. These datasets are processed into a simple and standardized format. A systematic benchmark with atomistic learning methods is presented, showcasing the value of using 3D atom-level data instead of 1D or 2D features. The authors argue that these datasets will serve as a stepping stone for machine learning researchers interested in developing methods for atomistic learning and rapidly advance this field. The paper also presents the best practices for each of the tasks, as well as the splitting and filtering criteria to ensure generalizability and reproducibility.

Strong points of the paper
- The authors present a systematic evaluation of atomistic learning across multiple tasks and show that 3D data consistently yields better performance than 1D and 2D methods.
- The experiments are thorough and the literature for every task has been covered extensively. The performance evaluation for each task is fair, respecting the corresponding metrics.
- Novel molecular tasks and their corresponding datasets are introduced. This is a strong point for this paper. The availability of standard curated datasets helps to advance the field by making it easier to develop and compare new methods for these tasks.

Weak points of the paper
- The idea of using atomistic learning or at least 3D derived features have already been implemented or at least contemplated in many of the presented tasks (Gilmer et al [1], Wu et al [2], Townshend et al. [3]). In fact, these methods are often SOTA in their fields. Because of this, I feel that there is not much novelty with atomistic learning.
- If the authors are selecting tasks where there are SOTA 3D methods, it would be also interesting to evaluate the performance against these methods (Townshend et al. [3] for instance). Or at least explain how their 3D approach is different.
- I feel that the selection of multiple tasks is limiting the authors in the amount of information that they can fit in the actual paper. As it is now, the paper relies heavily on the Appendix as important information is described there and not in the main paper.

My overall recommendation is a weak accept. This is because the methods used and the idea of atomistic learning are not novel. The novelty of the paper lies in the introduction of the curated datasets and the novel tasks defined by the authors. By creating a standardized set of prediction tasks and associated data sets, the authors have presented a resource that may help the community to compare 3D atomistic methods quickly and fairly. However, I personally feel that this work could be a better fit for a more biologically inclined venue.


Questions for the authors
- For the MSP task, could you explain why the problem was framed as a binary classification problem? There seems to be a fair amount of papers tackling the non-binary problem (Montanucci et al. [4], Cao et al. [5], Rodrigues et al. [6] to name a few).
- Is PPI the appropriate name for this task? Townshend et al., which as stated in the paper is a motivation for the approach, frames the problem as Protein Interface Prediction. The same goes for the baseline (Sanchez-Garcia et al. [7]). Although related, the name PPI could generate confusion as it spans a whole different literature.

Suggestions
- I feel that it would be interesting to have a more detailed discussion on why the atom-level data improves performance on each task.
- If what the authors want is to show that 3D features outperform 2D or 1D for these molecular tasks, then a systematic evaluation of the same methods but with different types of features would be more significant. For instance 3D CNNs vs 2D CNNs (where possible).

References
[1] Gilmer, J., Schoenholz, S. S., Riley, P. F., Vinyals, O., & Dahl, G. E. (2017). Neural message passing for quantum chemistry. arXiv preprint arXiv:1704.01212.
[2] Wu, Z., Ramsundar, B., Feinberg, E. N., Gomes, J., Geniesse, C., Pappu, A. S., ... & Pande, V. (2018). MoleculeNet: a benchmark for molecular machine learning. Chemical science, 9(2), 513-530.
[3] Townshend, R., Bedi, R., Suriana, P., & Dror, R. (2019). End-to-end learning on 3d protein structure for interface prediction. In Advances in Neural Information Processing Systems (pp. 15642-15651).
[4] Montanucci, L., Capriotti, E., Frank, Y., Ben-Tal, N., & Fariselli, P. (2019). DDGun: an untrained method for the prediction of protein stability changes upon single and multiple point variations. BMC bioinformatics, 20(14), 335.
[5] Cao, H., Wang, J., He, L., Qi, Y., & Zhang, J. Z. (2019). DeepDDG: predicting the stability change of protein point mutations using neural networks. Journal of chemical information and modeling, 59(4), 1508-1514.
[6] Rodrigues, C. H., Pires, D. E., & Ascher, D. B. (2018). DynaMut: predicting the impact of mutations on protein conformation, flexibility and stability. Nucleic acids research, 46(W1), W350-W355.
[7] Sanchez-Garcia, R., Sorzano, C. O. S., Carazo, J. M., & Segura, J. (2019). BIPSPI: a method for the prediction of partner-specific protein–protein interfaces. Bioinformatics, 35(3), 470-477.

---

> ### Author Response · Authors · 2020-11-24
> **Response to Reviewer 3**
>
> Thank you for your review; we are glad that the systematic and thorough nature of our evaluation comes through as a strong point of our paper. We respond to your concerns and questions below.
>
> > The idea of using atomistic learning or at least 3D derived features have already been implemented or at least contemplated in many of the presented tasks (Gilmer et al [1], Wu et al [2], Townshend et al. [3]). In fact, these methods are often SOTA in their fields. Because of this, I feel that there is not much novelty with atomistic learning.
>
> There are indeed several works which take into account 3D molecular features for some of the individual tasks we present. Works like Gilmer et al. and Townshend et al. are good examples of this, and their SOTA results provide evidence that a broader consideration of 3D atomic data will be a boon to the machine learning community. The unique value of ATOM3D is really the “opinionated” focus on 3D atomic data through a systematization of datasets, methods, and comparisons not only for small molecules, but also other biological macromolecules such as proteins and nucleic acids. Previous collections of datasets like MoleculeNet (Wu et al.) serve the distinct purpose of focusing on applications, not representations: they only have a couple of datasets at most that contain 3D information, and are mainly focused on small molecules.
>
> > If the authors are selecting tasks where there are SOTA 3D methods, it would be also interesting to evaluate the performance against these methods (Townshend et al. [3] for instance). Or at least explain how their 3D approach is different.
>
> Thank you for the suggestion. We have included performance metrics for SOTA 3D methods in the updated version of the manuscript where they exist (see Section C.4 and Table 7 in the Appendix), as well as clarified the novelty of our methods in Section 4. In most cases, our best-performing model matches or outperforms the SOTA 3D models: for RES, our 3DCNN implementation achieves an accuracy of 0.451 compared to 0.425 reported in Torng et al. [1] for a similar balanced training set; for MSP, our ENN and GNN both achieve higher AUROC (0.678 and 0.637, respectively) than the 3D-based method reported by Wang et al. [2] (0.569); for LEP, our 3DCNN achieves an AUROC of 0.824 compared to 0.770 for Glide, a widely-used 3D-based scoring program [3].
>
> The three datasets where SOTA outperforms our models are SMP, PIP, and LBAx. For PIP, our 3DCNN implementation is a replica of Townshend et al. [4] (the novelty is in the GNN and ENN methods, we have clarified this in the text), and it achieves an AUROC of 0.844; SOTA is the BIPSPI model that uses boosted decision trees not only on structural features, but also on sequence conservation features to achieve an AUROC of 0.919. For SMP, competing methods were deep learning networks applied to atomistic geometry, and no single method outperformed our ENN implementation on every SMP sub-task. For LBA, the SOTA (X-score) is an empirical regression-based scoring function based on structural features that has shown consistent performance on non-redundant datasets [5].
>
> > I feel that the selection of multiple tasks is limiting the authors in the amount of information that they can fit in the actual paper. As it is now, the paper relies heavily on the Appendix as important information is described there and not in the main paper.
>
> There is definitely a tradeoff between the comprehensiveness of the resource and the level of detail we can include in the main paper, but we decided to prioritize the availability of the datasets for all tasks. For the camera-ready version of the manuscript, we will elaborate on the dataset formatting details in the first two paragraphs of Appendix C.1 and move this and other important dataset information to the main text at the beginning of Section 3. Additionally, all dataset preparation details will be readily available in the Github repository to ensure that users can understand the datasets without reading the Appendix.
>
> > For the MSP task, could you explain why the problem was framed as a binary classification problem? There seems to be a fair amount of papers tackling the non-binary problem (Montanucci et al. [4], Cao et al. [5], Rodrigues et al. [6] to name a few).
>
> The binary version was created to match how such a network might be used in practice (i.e., determining if a proposed mutation improves over the unmutated version). You are correct that the non-binary version of this task is more common in the literature, and we were in fact already working on creating this version of the MSP dataset as well. We plan to have this ready along with the benchmark performance for all three methods by the camera-ready version.

---

> > ### Author Response · Authors · 2020-11-24
> > **Reponse to Reviewer 3 - Continued**
> >
> > > Is PPI the appropriate name for this task? Townshend et al., which as stated in the paper is a motivation for the approach, frames the problem as Protein Interface Prediction. The same goes for the baseline (Sanchez-Garcia et al. [7]). Although related, the name PPI could generate confusion as it spans a whole different literature.
> >
> > Recognizing that PPI is an overloaded term, we have changed the acronym to PIP in the manuscript.
> >
> > > Suggestions
> > > I feel that it would be interesting to have a more detailed discussion on why the atom-level data improves performance on each task.
> > > If what the authors want is to show that 3D features outperform 2D or 1D for these molecular tasks, then a systematic evaluation of the same methods but with different types of features would be more significant. For instance 3D CNNs vs 2D CNNs (where possible).
> >
> > These are definitely good suggestions, and items we will consider moving forward!
> >
> > References:
> >
> > [1] Torng, W., & Altman, R. B. (2017). 3D deep convolutional neural networks for amino acid environment similarity analysis. BMC Bioinformatics, 18(1), 302.
> >
> > [2] Wang, M., Cang, Z., & Wei, G.-W. (2020). A topology-based network tree for the prediction of protein–protein binding affinity changes following mutation. Nature Machine Intelligence, 2(2), 116–123.
> >
> > [3] Friesner, R. A., Banks, J. L., Murphy, R. B., Halgren, T. A., Klicic, J. J., Mainz, D. T., … Shenkin, P. S. (2004). Glide: A New Approach for Rapid, Accurate Docking and Scoring. 1. Method and Assessment of Docking Accuracy. Journal of Medicinal Chemistry, 47(7), 1739–1749.
> >
> > [4] Townshend, R., Bedi, R., Suriana, P., & Dror, R. (2019). End-to-end learning on 3d protein structure for interface prediction. Advances in Neural Information Processing Systems (pp. 15642-15651).
> >
> > [5] Li, Y., & Yang, J. (2017). Structural and sequence similarity makes a significant impact on machine-learning-based scoring functions for protein–ligand interactions. Journal of Chemical Information and Modeling, 57(4), 1007-1012.

---

### Official Review · AnonReviewer2 · 2020-10-29
**Main contribution is the dataset curation**

**Rating:** 5
**Confidence:** 3

**Review:**

This paper is concerned with 3D molecule learning. They propose a collection of existing and new datasets (curated from existing datasets). They show that a lot of existing tasks can do well when 3D structure is considered.

The main contribution of the paper is in curating the datasets into a well defined framework with consistent splits and evaluation metrics. This would allow the community members to easily benchmark their approaches against a variety of tasks. Also, the tasks encompass a range of modalities including RNA, Proteins and Small molecules. Some tasks like Ligand Binding Affinity (LBA) and Ligand Efficacy Prediction (LEP) requires modeling representations of both proteins and small molecules. I believe this benchmark dataset will be useful to the community.

I have some complaints with the experimental evaluations. For SMPs, a variety of unsupervised feature extractions methods have been proposed, which can be applied on 1D (SMILES representations) as well as graphs. For example see [1] and [2]. In QM9, N-Gram XGB (See table 2 from [2]), performs very well (top 1 performance on 9 out of 12 tasks). Did adding 3D information improve on this result? It is not clear.

Similarly, for binding affinity prediction, the authors compare with DeepDTA. However, DeepAffinity[3] has shown promising results just using 1D representations. Are 3D models better than this? What happens if we use the same models on PDBBind data?

[1]  Rong et al: GROVER: Self-supervised Message Passing Transformer on Large-scale Molecular Data
 https://arxiv.org/pdf/2007.02835.pdf
[2] Liu et al: N-Gram Graph: Simple Unsupervised Representation for Graphs, with Applications to Molecules
[3] Karimi et al: DeepAffinity: Interpretable Deep Learning of Compound-Protein Affinity through Unified Recurrent and Convolutional Neural Networks

---

> ### Author Response · Authors · 2020-11-24
> **Response to Reviewer 2**
>
> Thank you for your comments, and we are glad you agree that this benchmark suite will be useful to the community!  We respond to your individual concerns below.
>
> > In QM9, N-Gram XGB performs very well (top 1 performance on 9 out of 12 tasks). Did adding 3D information improve on this result?
>
> N-Gram XGB [1] is indeed a relevant model to compare against, and we find that adding 3D information does improve on this result. Specifically, to evaluate N-Gram XGB’s performance, we trained and tested it on the SMP (QM9) dataset and obtained the following results using the same split, tasks and units as in our paper (note that some of the units deviate from [1] and thermochemical energy is subtracted from U0, U298, H298, and G298 here):
> * mu [D]:   0.520 +/- 0.010
> * alpha [bohr^3]:   0.593 +/- 0.016
> * homo [eV]:   0.130 +/- 0.003
> * lumo [eV]:   0.137 +/- 0.004
> * gap [eV]:   0.184 +/- 0.007
> * r2 [bohr^2]:  59.285 +/- 2.456
> * zpve [meV]:  12.156 +/- 0.344
> * cv [cal/mol/K]:   0.333 +/- 0.008
> * u0_atom [eV]:   0.218 +/- 0.007
> * u298_atom [eV]:   0.220 +/- 0.007
> * h298_atom [eV]:   0.220 +/- 0.006
> * g298_atom [eV]:   0.215 +/- 0.006
>
> This performance is worse than our GNN and ENN methods and slightly worse than our current non-3D baseline (Tsubaki et al. 2019 [2], see Tables 3 and 8). We have added these numbers to Table 3, and inserted a reference to N-Gram XGB in the related work section.
>
> > DeepAffinity has shown promising results just using 1D representations. Are 3D models better than this? What happens if we use the same models on PDBBind data?
>
> We agree this is also a good baseline.  We are still working to run DeepAffinity [3] on PDBBind data, and will update here if we manage to get this running before the deadline!
>
> References:
>
> [1] Liu, S., Demirel, M. F. & Liang, Y. (2019). N-Gram Graph: Simple Unsupervised Representation for Graphs, with Applications to Molecules. Advances in Neural Information Processing Systems 32, pp. 8466-8478
>
> [2] Tsubaki, M., Tomii, K. & Sese J. (2019). Compound-protein interaction prediction with end-to-end learning of neural networks for graphs and sequences. Bioinformatics 35(2): 309-318
>
> [3] Karimi, M., Wu, D., Wang, Z., Shen, Y. (2019). DeepAffinity: interpretable deep learning of compound-protein affinity through unified recurrent and convolutional neural networks. Bioinformatics 35(18):3329-3338

---

> > ### Comment · AnonReviewer2 · 2020-11-25
> > **Additional baselines**
> >
> > Thank you for adding the additional baseline.

---

> > ### Author Response · Authors · 2020-11-25
> > **Additional baseline for ligand binding affinity (LBA)**
> >
> > On our LBA dataset, DeepAffinity [3] achieves the following performance:
> > - RMSE:   1.893 +/- 0.119
> > - Pearson R:   0.415 +/- 0.088
> > - Spearman R:   0.426 +/- 0.097
> >
> > This is worse than our current non-3D baseline. Compared to 3D methods, it is only competitive with the GNN on the RMSD (though not on the Pearson or Spearman correlation coefficients).  We add the DeepAffinity performance to Table 5 and include detailed information on this baseline to Appendix C.3.
> >
> > We also note that our LBA dataset is derived directly from PDBBind, with the main difference being the conversion to our unified data representation and the choice of a sequence identity–based splitting criterion in place of the less stringent PDBBind core dataset. See Appendix B.1 for further details on the derivation of the ligand binding affinity dataset.

---

### Author Response · Authors · 2020-11-24
**Thank you to the reviewers**

We thank the reviewers for taking the time to provide such detailed feedback and thoughtful suggestions.  We address each of the reviewer’s questions and concerns individually.  We have also uploaded an updated copy of the main and supplementary text with the revisions we mention below.

---

### Decision · Program_Chairs · 2021-01-07
**Final Decision**

**Decision:**

Reject

**Comment:**

For many problems such as ligand-protein binding, quantitative structure activity prediction (QSAR), predicting protein function from structure, etc., the 3D geometry of the molecules is of great importance.  One way to represent this is simply to assign locations to all atoms in 3-dimensional space.  If using graph convolutional kernels or other relational representations such that aligning molecules is not necessary, these approaches with 3D geometry can be efficient and far more effective than 1D or 2D representations.  The contribution of the paper is to make this point and to produce a resource with this kind of 3D data.  Such a resource would be of high value.  Nevertheless, reviewers feel provision of such a resource is perhaps not a major contribution to the ICLR and ML communities.  There is a sense that more innovative and substantial contribution would come from addressing also the challenge that 3D geometry can changes and that there may be multiple low-energy conformations of biomolecules that should be considered.  The authors contend that unlike ligands which are small and may have many low-energy conformations, large biomolecules have a much more constrained conformational space.

This meta-reviewer is sympathetic to the authors' point and appreciates the importance of the resource.  Nevertheless, even large biomolecules often have some portions of flexible conformation and high 3D structure variation that should be considered.  And indeed addressing the kind of multiple instance problem that arises by considering multiple conformations of large molecules or of ligands binding to large molecules would certainly require and likely yield bigger ICLR/ML innovations.  In the end the paper contributes a useful resource but does not excite the reviewers substantially enough, without those extensions or others, for a recommendation of acceptance at this time.